# Community Knowledge about Attention Deficit Hyperactivity Disorder in Saudi Arabia: A Cross-Sectional Study

**DOI:** 10.3390/healthcare11010054

**Published:** 2022-12-25

**Authors:** Amal Khaleel Abu Alhommos, Fatimah Mohammed AlHadab, Rwan Adel Alalwan, Sara Tawfiq Alabduladhem, Zahraa Ali Alnaser, Sarah Saad Alnami

**Affiliations:** Pharmacy Practice Department, Clinical Pharmacy College, King Faisal University, Alhasa 43518, Saudi Arabia

**Keywords:** attention deficit hyperactivity disorder, community, knowledge, public, Saudi Arabia

## Abstract

**Objectives**: The majority of earlier studies on knowledge and attitudes around ADHD have been focused on parents of children with ADHD, the child themselves, primary care providers, teachers, and children’s families, and they have found that they have negative perceptions of ADHD. This study aimed to explore community knowledge about ADHD in Saudi Arabia. **Methods**: A cross-sectional study using an online survey was conducted in Saudi Arabia in January 2022 to explore community knowledge about ADHD in Saudi Arabia. The convenience sampling technique was used to identify eligible participants and invite them to take part in the study. Logistic regression analysis was used to identify ADHD knowledge predictors. **Results**: A total of 718 participants were involved in this study. The participants in our study showed a weak level of knowledge about ADHD with a mean score of 6.4 (SD: 2.2) out of 17 (which is equal to 37.6% out of the maximum obtainable score). Males, those who work outside the healthcare sector, and those who are retired were less likely to be knowledgeable about ADHD compared to others (*p* ≤ 0.05). At the same time, participants who reported that they work inside the healthcare sector were more likely to be knowledgeable about ADHD compared to others (*p* ≤ 0.01). **Conclusions**: Our study showed that there is insufficient public knowledge about ADHD. The development of educational interventions is necessary to raise public awareness of ADHD. Social media platforms can be used to deliver education campaigns. All members of the community, including parents and those who want to get married, should be the target of these initiatives.

## 1. Introduction

One of the most common psychological disorders is attention deficit hyperactivity disorder (ADHD), which affects between 5% and 8% of children worldwide. The majority of those affected are boys, and it frequently lasts into adulthood [1,2,3]. The prevalence rate of ADHD in children is high in the Kingdom of Saudi Arabia (KSA) [4,5], where the prevalence rate is 16.4% among male children in elementary school, and ranges between 11.6% and 13.5% among male and female children [6,7,8,9].

The three main symptoms of ADHD are (a) hyperactivity (excessive talking, tapping, fidgeting, or excessive movement that is not appropriate for the situation); (b) inattention (inability to focus); and (c) impulsivity (acting hastily and without thinking, possibly harming others) [3,10]. The child’s performance in school and in daily activities is impacted by ADHD [3]. Additionally, children with ADHD have higher rates of comorbid psychiatric illnesses and are frequently admitted to hospitals, which results in higher total medical costs for them than for children without ADHD [11,12].

The majority of earlier studies on knowledge and attitudes around ADHD have been focused on parents of children with ADHD, the child themselves, primary care providers, teachers, and children’s families, and they have found that they have negative perceptions of ADHD [13,14,15,16,17,18,19,20,21]. The perspectives of the larger culture about ADHD, however, have received little attention in prior studies [22,23,24]. To improve outcomes for people with ADHD through support, treatment, and an accurate diagnosis, positive attitudes and accurate knowledge about ADHD within the larger community are crucial [22,23].

In the KSA, many studies have found an insufficient level of knowledge about ADHD, and this was evaluated among the following groups: a sample of teachers in the Al-Rusaifah district in Makkah city [25], female teachers at elementary schools in Jeddah [26], primary health care physicians in the Aseer Region [27], and medical students at the Faculty of Medicine at King Abdulaziz University in Jeddah [28]. However, another previous study found that two-thirds of male primary school teachers in Riyadh had knowledge about ADHD [29]. There are no previous studies evaluating knowledge and attitudes toward ADHD within the community in KSA. Therefore, this study aimed to explore community knowledge about ADHD in Saudi Arabia.

## 2. Methods

### 2.1. Study Design

A cross-sectional study using an online survey was conducted in Saudi Arabia in January 2022 to explore community knowledge about ADHD in Saudi Arabia.

### 2.2. Sampling Strategy and Study Population

The convenience sampling technique was used to identify eligible participants and invite them to take part in the study. This sampling technique falls under the category of non-probability sampling. This study included eligible people who satisfied our inclusion criteria and were available to participate in the study. Social media platforms (Facebook and WhatsApp) were used to reach out to the general public and invite them to take part in this study. We have distributed the study link and the initiation letter through various Facebook pages and WhatsApp groups that are relevant to the general public and not only for a specific group in order to reach the Saudi Arabian population as a whole.

All participants gave their informed consent voluntarily and were therefore exempt from providing written consent. At the beginning of the survey, the aims and objectives of the study were presented in detail. The participants received no compensation.

Participants who were at least 18 years old and were currently living in Saudi Arabia met the inclusion criteria. If a participant was under 18 years old or could not read or understand Arabic, they were not allowed to participate.

### 2.3. Study Tool

An extensive literature review served as the foundation for the study tool’s development. There were three sections with a total of 27 questions. The participants’ demographics were covered in the first section, which included five questions concerning age, gender, education level, residence, and employment status. The second section of the questionnaire consisted of 17 multiple-choice questions (MCQ) that probed participants’ knowledge about ADHD. Each correct response was given a weight of one, while each incorrect response was given a weight of zero. The maximum score was 17, with a higher score denoting more knowledge. Participants’ knowledge of the risk factors, symptoms, consequences, and treatment options for ADHD was tested by knowledge questions. The third section of the questionnaire consisted of five multiple-choice questions (MCQs) that asked the participants about their knowledge of ADHD, whether they knew anyone in their family who had the condition, how much they knew about its treatment, whether they knew which centers in their area managed ADHD, and where they got their information about the condition.

### 2.4. Statistical Analysis

The demographic characteristics of the participants were described using descriptive statistics. For variables that were normally distributed, continuous data were reported as mean ± SD. Categorical data were reported as percentages (frequencies). The odds ratios (ORs) and 95% confidence intervals (CIs) for individuals who are more likely to be knowledgeable about ADHD were calculated using logistic regression. The cut-off for the logistic regression was based on the mean knowledge score of the study participants. The mean knowledge scores for various demographic groups were compared using an independent samples t-test and a one-way analysis of variance (ANOVA). The statistical analyses were carried out using SPSS (version 27).

## 3. Results

### 3.1. Participants Demographic Characteristics

A total of 718 participants were involved in this study. The majority of the study participants (73.4%) were females. Around one-third of the study participants (28.4%) were aged 18–25 years. The vast majority of the study participants (92.8%) were residents of the eastern area. Around one-third of the study participants (35.4%) reported that they were working in the healthcare sector. More than half of the study participants (68.2%) reported that they held a bachelor’s degree. A similar percentage of them (61.0%) reported that they knew some information about ADHD. Around one-fifth of the study participants (20.2%) reported that they have a family member diagnosed with ADHD. Only 10.3% of the study participants reported that they had good information about the treatment of ADHD. Around one-fifth of the study participants (18.4%) reported that they know centers specialized in treating ADHD in their area. Around half the study participants (51.5%) reported that social media is their main source of information about ADHD. Table 1 describes the demographic characteristics of the study participants.

### 3.2. Knowledge about Attention Deficit Hyperactivity Disorder

The participants in our study showed a weak level of knowledge about ADHD with a mean score of 6.4 (SD: 2.2) out of 17 (which is equal to 37.6% out of the maximum obtainable score). The mean ADHD knowledge score showed a statistically significant difference between participants from different demographic groups based on their gender, age, employment status, education and having family member diagnosed with ADHD (*p* ≤ 0.01). Females, young participants (aged 18–25 years), those who work inside the healthcare sector, those with higher level of education and those who have a family member diagnosed with ADHD had a higher knowledge score about ADHD compared to others. Table 2 presents the mean ADHD knowledge score stratified by demographic characteristics.

### 3.3. Factors Affecting Participants’ Knowledge about ADHD

Binary logistic regression analysis identified that males, those who work outside the healthcare sector, and those who are retired were less likely to be knowledgeable about ADHD compared to others (*p* ≤ 0.05). At the same time, participants who reported that they work inside the healthcare sector were more likely to be knowledgeable about ADHD compared to others (*p* ≤ 0.01) (Table 3).

## 4. Discussion

The key findings of our study are: (1) only 10.3% of the study participants reported that they have good information about the treatment of ADHD, (2) around one-fifth the study participants reported that they know centers specialized in treating ADHD in their area, (3) around half the study participants reported that social media is their main source of information about ADHD, (4) the participants in our study showed a weak level of knowledge about ADHD, (5) males, those who work outside the healthcare sector, and those who are retired were less likely to be knowledgeable about ADHD compared to others, and (6) participants who reported that they work inside the healthcare sector were more likely to be knowledgeable about ADHD compared to others.

Only 10.3% of our study participants reported that they have good information about the treatment of ADHD. The participants in our study showed a weak level of knowledge about ADHD with a mean score of 6.4 (SD: 2.2) out of 17 (which is equal to 37.6% out of the maximum obtainable score). This confirmed the findings of a previous study that was conducted in Indonesia by Murtani et al. [24], which reported that community members had poor levels of knowledge and understanding regarding ADHD, accounting for 56.8% of the respondents. However, it was lower than the findings of other studies that explored the knowledge level among parents and teachers [30,31,32,33]. A previous study that was conducted in KSA and explored the knowledge of teachers about ADHD reported a higher level of knowledge (90.0%) [34]. Murtani et al. justified the weak knowledge found due to the variation in the respondents’ demographic backgrounds, such as ethnicity [24]. In our study, around one-fifth of the study participants reported that they have a family member with ADHD. This is one of the main contributing factors that could justify the low level of knowledge. Previous literature reported that having relatives or a family member with ADHD increases the individual knowledge and experience with the disease [30,35]. According to a previous survey study in the United States (US), the majority of the general public did not know the symptoms of ADHD [36]. Another previous study in Iran found that only 45.3% of parents of children with ADHD were aware that the disorder’s symptoms can be managed with medication, that 71.3% thought their child’s hyperactivity helped the disorder by releasing energy, and that 52.7% thought psychological testing was required to make the diagnosis [37]. Understanding ADHD is important, especially for parents, as it aids them in how they raise their affected children. This includes understanding how the disorder is diagnosed, how ADHD treatment affects affected children’s daily lives, and how to improve treatment compliance [30].

Around half the study participants (51.5%) reported that social media is their main source of information about ADHD. This supported the findings of a prior study carried out in Saudi Arabia, which found that the internet and social media were the primary sources of information about ADHD [29]. Interestingly, social media is believed to have a beneficial influence in the Gulf region. Relying on non-medical sources of information as the main reference for medical information about ADHD is a common problem that was reported in previous literature [34]. Relying on reliable sources is important to avoid misconceptions about ADHD. Decision-makers are therefore urged to encourage the development of a particular course on ADHD for various community segments, and education should be included in faculties’ training curriculum. Given that these were the most popular information sources, broadcasting these educational programs on different social media platforms may be quite beneficial.

Males, those who work outside the healthcare sector, and those who are retired were less likely to be knowledgeable about ADHD compared to others. A previous study in Taiwan reported that housewives spend more time controlling their kids’ behavior, and as they gain parenting experience, they become more knowledgeable about ADHD [30]. Similarly, a previous study in Korea has found that female teachers were more knowledgeable about ADHD compared to male teachers [38]. Our study did not find an association between education level or age and knowledge about ADHD. This aligned with the findings of previous literature in other Middle Eastern countries, which concluded that there is a general lack of ADHD awareness in the Gulf region [39].

There are many advantages to this study. The fact that this is one of the few research studies among Middle Eastern Arabic-speaking countries to look into public knowledge of ADHD increases the generalizability of the results. However, there are some limitations. Our ability to determine causation between research variables was constrained by the study’s methodology, a cross-sectional survey design. Our inability to compare our results with those of Arabic-speaking countries with a comparable environment and culture was due to the fact that no earlier research had been performed on the general population exclusively, as the subjects of earlier studies were parents and teachers. We were not able to estimate the response rate for our study as we employed a convenience sampling technique, which might lead to non-response bias. The majority of the participants in this study were from the Eastern region, which restricts the generalizability of the study findings. Despite this, we assume that the findings of this study are of huge value due to the important characteristics of the Eastern region. In 2017, the Eastern region had a population of 4,900,325 people. The Eastern region is the most eastern of Saudi Arabia’s 13 provinces. It is the province with the biggest land area and the third most inhabitants in Saudi Arabia [40]. In addition, more than half of the participants were females, which might have affected the generalizability of our study findings. This is one of the limitations of online survey studies. Perhaps because they are a simple, practical, and affordable method of data collecting, online surveys are becoming more popular and commonly used. Therefore, our findings should be interpreted carefully.

## 5. Conclusions

Our study showed that there is insufficient public knowledge of ADHD. The development of educational interventions is necessary to raise public awareness of ADHD. Social media platforms can be used to deliver education campaigns. All members of the community, including parents and those who want to get married, should be the target of these initiatives.

## Figures and Tables

**Table 1 healthcare-11-00054-t001:** Demographic characteristics of the study participants.

Demographic Variable	Frequency	Percentage
**Gender**		
Female	527	73.4%
**Age category**		
18–25 years	204	28.4%
26–35 years	128	17.8%
36–45 years	184	25.6%
46–55 years	139	19.4%
55 years and above	63	8.8%
**Area of residency**		
Central area	17	2.4%
Eastern area	666	92.8%
Western area	11	1.5%
Southern area	13	1.8%
Northern area	11	1.5%
**Employment status**		
Work outside the healthcare area	254	35.4%
Unemployed	213	29.7%
Work inside the healthcare area	75	12.8%
Student	91	12.7%
Retired	85	11.8%
**Education**		
Secondary school level or lower	195	27.2%
Bachelor’s degree	490	68.2%
Higher education	33	4.6%
**How much do you know about attention deficit hyperactivity disorder (ADHD)?**		
I don’t know anything about it	104	14.5%
I know some information about it	438	61.0%
I have good information about it	176	24.5%
**Do you have any family member with ADHD?**		
Yes	145	20.2%
**How much do you know about the treatment of ADHD?**		
I don’t know anything about it	340	47.4%
I know some information about it	304	42.3%
I have good information about it	74	10.3%
**Do you know what centers specialized in treating ADHD in your area?**		
Yes	132	18.4%
**What are your sources of information about ADHD?**		
Social media	370	51.5%
Internet	246	34.3%
Family	176	24.5%
Books and journals	126	17.5%
Friends	113	15.7%
Television	101	14.1%

**Table 2 healthcare-11-00054-t002:** Mean knowledge score stratified by demographic characteristics.

Demographic Variable	Mean (SD)	*p*-Value
**Gender**		
Female	6.6 (2.1)	**0.000 *****
Male	5.9 (2.5)	
**Age category**		
18–25 years	6.8 (2.1)	**0.001 ****
26–35 years	6.7 (2.2)	
36–45 years	6.1 (2.1)	
46–55 years	6.0 (2.5)	
55 years and above	5.9 (2.4)	
**Area of residency**		
Central area	6.3 (2.7)	0.666
Eastern area	6.4 (2.2)	
Western area	6.1 (2.8)	
Southern area	7.0 (1.7)	
Northern area	5.6 (1.9)	
**Employment status**		
Work inside the healthcare area	7.5 (2.4)	**0.000 *****
Work outside the healthcare area	6.2 (2.2)	
Unemployed	6.3 (1.9)	
Student	6.9 (2.1)	
Retired	5.7 (2.6)	
**Education**		
Secondary school level or lower	6.0 (2.1)	**0.002 ****
Bachelor’s degree	6.5 (2.2)	
Higher education	7.3 (2.5)	
**Do you have any family members with ADHD?**		
No	5.8 (2.4)	**0.000 *****
Yes	6.5 (2.1)	

** *p* ≤ 0.01; *** *p* ≤ 0.001.

**Table 3 healthcare-11-00054-t003:** Binary logistic regression analysis.

Demographic Variable	Odds Ratio	95% CI
**Gender**		
Female (Reference group)	1.00	
Male	**0.58**	**0.41–0.81 ****
**Age category**		
18–25 years (Reference group)	1.00	
26–35 years	1.28	0.87–1.88
36–45 years	0.73	0.52–1.03
46–55 years	0.79	0.54–1.14
55 years and above	0.74	0.44–1.25
**Employment status**		
Student (Reference group)	1.00	
Work inside the healthcare area	**2.31**	**1.40–3.82 ****
Work outside the healthcare area	**0.72**	**0.53–0.99 ***
Unemployed	1.05	0.76–1.45
Retired	**0.59**	**0.37–0.95 ***
**Education**		
Secondary school level or lower (Reference group)	1.00	
Bachelor’s degree	1.33	0.97–1.83
Higher education	1.40	0.70–2.83
**Do you have any family members with ADHD?**		
No (Reference group)	1.00	
Yes	0.87	0.67–0.92

* *p* ≤ 0.05; ** *p* ≤ 0.01.

## Data Availability

Data are available on request from the corresponding author.

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
