# Peer review of "Community Knowledge about Attention Deficit Hyperactivity Disorder in Saudi Arabia: A Cross-Sectional Study"

_healthcare, 2022, doi:10.3390/healthcare11010054_

Round 1
Reviewer 1 Report
Thank you for giving me the precious opportunity to review this article. This study investigated the knowledge about ADHD in the community in Saudi Arabia. The study would alter the attitude around ADHD and subsequently contribute to decreasing negative biases toward people with ADHD in the Saudi community. The methodology used in this study is appropriate, however, there are some points to be clarified in this paper.
Over 90 % of the participants is living in the Eastern area. Could authors describe the characteristics of this area for those who live in the other part of the world?
Over 70 % of the participants are female. Could authors speculate this cause and the effect of this gender ratio on the results?
Over 20 % of participants have family history of ADHD which is much higher than the figure in general population. This must have huge impact on the results.
Is any compensation paid to the participants?
Author Response
Thank you for giving me the precious opportunity to review this article. This study investigated the knowledge about ADHD in the community in Saudi Arabia. The study would alter the attitude around ADHD and subsequently contribute to decreasing negative biases toward people with ADHD in the Saudi community. The methodology used in this study is appropriate, however, there are some points to be clarified in this paper.
- First of all, we would like to thank the reviewer for the time and efforts in reviewing our paper.
Over 90 % of the participants is living in the Eastern area. Could authors describe the characteristics of this area for those who live in the other part of the world?
- Thank you for this comment. We have now addressed this comment in page 6, lines 203-208.
Over 70 % of the participants are female. Could authors speculate this cause and the effect of this gender ratio on the results?
- Thank you for this comment. We totally agree with the reviewer on this point and this is one of the limitation of online survey studies. We have now highlighted this point further in the limitations section, lines 209-213.
Over 20 % of participants have family history of ADHD which is much higher than the figure in general population. This must have huge impact on the results.
- Thank you for this comment. We agree with the reviewer on this point, however, we assume that this was because of the fact that participants with family history of ADHD would be more interested in participation in such type of studies as they would be more interested than other people.
Is any compensation paid to the participants?
- Thank you for this comment. No compensation was provided to the participants. We have now added this point to the method section, line 74.
Reviewer 2 Report
The main question of this study is important, though its interest is more locally specific than international.
The results are clear and well written, significant, and driven to practical implications. Therefore, I believe this study deserves to be published. First, however, I have several comments:
1. This study looks at the community as a whole. However, the population recruitment method is unclear, and the population is not defined enough.
What population was approached by the social media? Through what channels (private, hospital's)?
How come so many of the responders are from the healthcare community? Why is there such a large percentage of people from the Eastern part of KSA?
These questions call for bias since it raises the question of the socio-economical levels that had access to the study. Thus, they might not reflect the community at large.
I believe there should be a better description of the population selection methods and a paragraph about the limitations resulting from the possible biases described above.
Author Response
The main question of this study is important, though its interest is more locally specific than international. The results are clear and well written, significant, and driven to practical implications. Therefore, I believe this study deserves to be published. First, however, I have several comments:
- First of all, we would like to thank the reviewer for the time and efforts in reviewing our paper.
- This study looks at the community as a whole. However, the population recruitment method is unclear, and the population is not defined enough.
- Thank you for this comment. The study participants were invited using convenience sampling technique, which involved inviting them through social media platforms, see lines 66-68. We have now added further details on the population recruitment technique used in this study, see lines 64-66. The study population were participants who were at least 18 years old and were currently living in Saudi Arabia met the inclusion criteria. If a participant was under 18 years old or could not read or understand Arabic, they were not allowed to participate, see lines 75-77.
What population was approached by the social media? Through what channels (private, hospital's)?
- Thank you for this comment. In order to reach the general public in Saudi Arabia we have distributed the study link along with the initiation letter through different Facebook pages and WhatsApp groups that are of interest to the general public and not just for a specific group, we have now added these details in lines 68-71.
How come so many of the responders are from the healthcare community? Why is there such a large percentage of people from the Eastern part of KSA?
- Thank you for this comment. We have now addressed this comment and highlighted this point in the study limitations in page 6, lines 203-208.
These questions call for bias since it raises the question of the socio-economical levels that had access to the study. Thus, they might not reflect the community at large.
- Thank you for this comment. We totally agree with the reviewer and have now addressed this comment and highlighted this point in the study limitations in pages 6-7, lines 203-213.
I believe there should be a better description of the population selection methods and a paragraph about the limitations resulting from the possible biases described above.
- Thank you for this comment. We have now addressed the reviewer comments and highlighted the study limitations in lines 196-213.